Predicting the influence of extreme temperatures on grain production in the Middle-Lower Yangtze Plains using a spatially-aware deep learning model

Mu Zijun 1
Xia Junfei jxia83@gatech.edu 2 3
1 Nanjing Smardaten Technologies Co., Ltd , Nanjing , Jiangsu , China
2 Georgia Institute of Technology , Atlanta , GA , United States of America
3 Rosenstiel School of Marine, Atmospheric and Earth Science, University of Miami , Miami , FL , United States of America
Phairuang Worradorn
Electronic publication date: 2024 Oct 18
Publication date: 2024
Volume: 12
Electronic Location ID: e18234
Received 2024 May 24; Accepted 2024 Sep 13
Copyright: ©2024 Mu and Xia
Copyright year: 2024
Copyright holder: Mu and Xia
License: This is an open access article distributed under the terms of the Creative Commons Attribution License, which permits unrestricted use, distribution, reproduction and adaptation in any medium and for any purpose provided that it is properly attributed. For attribution, the original author(s), title, publication source (PeerJ) and either DOI or URL of the article must be cited.
License URL: https://creativecommons.org/licenses/by/4.0/

Keywords: Extreme temperature event, NEX-GDDP-CMIP6, Grain production model, Convolutional autoencoder, Random forest, Shared socioeconomic pathway, Yangtze river basin, Sustainable agriculture, Climate change

Funding: The authors received no funding for this work.

==============================
Grain crops are vulnerable to anthropogenic climate change and extreme temperature events. Despite this, previous studies have often neglected the impact of the spatio-temporal distribution of extreme temperature events on regional grain outputs. This research focuses on the Middle-Lower Yangtze Plains and aims to address this gap as well as to provide a renewed projection of climate-induced grain production variability for the rest of the century. The proposed model performs significantly superior to the benchmark multilinear grain production model. By 2100, grain production in the MLYP is projected to decrease by over 100 tons for the low-radiative-forcing/sustainable development scenario (SSP126) and the medium-radiative-forcing scenario (SSP245), and about 270 tons for the high-radiative-forcing/fossil-fueled development scenario (SSP585). Grain production may experience less decline than previously projected by studies using Representative Concentration Pathways. This difference is likely due to a decrease in coldwave frequency, which can offset the effects of more frequent heatwaves on grain production, combined with alterations in supply-side policies. Notably, the frequency of encoded heatwaves and coldwaves has a stronger impact on grain production compared to precipitation and labor indicators; higher levels of projected heatwaves frequency correspond with increased output variability over time. This study emphasizes the need for developing crop-specific mitigation/adaptation strategies against heat and cold stress amidst global warming.

Introduction

Heatwave-related damage may wipe out the equivalent of 4% of Africa’s GDP in 2022 (Romanello et al., 2023). Exacerbated by climate change, extreme temperature events (ETEs), such as heatwaves (HWs) and cold waves (CWs), also have the potential to impede crop growth and development: intense HWs negatively affect phase development of grain crops such as rice and maize (Sánchez, Rasmussen & Porter, 2013), and frost damage from CWs can influence the beginning times of crop phenology (Mijnsbrugge et al., 2021). Under increasingly extreme climate conditions, precisely predicting crop output variability is crucial for ensuring future food security and mitigating economic losses.

In the changing climate, predicting crop yields/outputs involves evaluating their correlations with endogenous factors, such as phenotype, labor, and technology (Jung & Sim, 2018); exogenous factors, such as management and climate (Enovejas et al., 2020); and the interactions between these factors. Among algorithms, the Random Forest (RF) has been widely used in predicting yield/output of various crops. Khan, Li & Maimaitijiang (2022) proposed a geographically weighted RF regression approach to predict corn production in the US Corn Belt; Fradgley et al. (2023) showed that RF outperformed simplified models in grain output prediction; and Chen, Zhou & Ren (2024) established a RF model for grain yield estimation of wheat-maize rotation cultivated land, achieving high accuracy of 93.4%. However, previous studies have not focused heavily on the effect of prolonged heat/cold stress (therefore, have neglected the impact of extreme temperature events) and prioritized assessing temporal patterns over the spatial distribution of climate data.

Moreover, a majority of the current literature concerning grain productivity utilizes climate projections from general circulation models (GCMs) to project future climate-induced crop production variations. However, previous studies such as Chen et al. (2020), Sun et al. (2023) and Lafferty et al. (2021) primarily utilized Coupled Model Intercomparison Project Phase 5 (CMIP5), which has been outcompeted by the latest CMIP6 both on climatological temperature (Jiang et al., 2020) and temperature extremes (Fan et al., 2020), especially in China. In CMIP6, the Scenario Model Intercomparison Project (ScenarioMIP) assesses the range of model responses to more holistic Shared Socioeconomic Pathway (SSP) scenarios via adjusted Representative Concentration Pathway (RCP, from CMIP5) forcings, as well as incorporating land use changes (not included in CMIP5) (Bond & Scott, 2022). Furthermore, Thrasher et al. (2022) recently published a downscaled dataset for CMIP6 model outputs, which has enabled better regional-scale application of CMIP6 global projections.

A major agriculture hub in Eastern China, the Yangtze River Basin (hereafter referred to as the YRB) is also particularly influenced by climate change and its associated climate extremes. Extreme heat has triggered a 5–10% decline in rice production in the Jiangxi and Hunan provinces in the YRB (Fu et al., 2023). These provinces are parts of the Middle and Lower reaches of the YRB, better known as the Middle-Lower Yangtze Plains (MLYP). Observing the MLYP’s significant and representative reaction to climate change, this area will constitute this study’s primary area of focus.

This study contributes to improved accuracy in the projection of climate-induced grain production variability in the MLYP. To achieve this, this study proposes an novel model that (1) combined temporal and often-neglected spatial attributes of (2) both climate averages and climate extremes. This study also (3) updates the impact of future climate change with the projections of the latest downscaled CMIP6 model ensemble.

With the innovative model, this study has a tri-fold goal: (1) develop a space-aware representation of grain production in a warming climate through a high-performance model, (2) analyze the impact of ETEs on grain production in the MLYP provinces, and (3) use state-of-the-art climate projections to compare grain production under different SSP scenarios. Findings can provide insights for better policy decisions and civil preparations for increasing severity and frequency of ETEs, and model results from this study can help inform climate adaptation and mitigation to achieve a more innovative and more sustainable system of agriculture in the age of climate change.

Data and Materials

Defining Grain Crops.

According to National Bureau of Statistics of P. R. China (2023), the administrative definition of grain crops include rice, wheat, corn, millet, sorghum, and “other grains” (such as barley, oats, buckwheat, etc.). We used grain data from official statistical yearbooks conforming to this definition.

Area of study

Based on examples of previous studies (such as Ding et al. (2017)), the border of the geographical Middle-Lower reaches of the Yangtze River Basin (YRB), and secondary (province-level) administrative boundaries, the Middle-Lower Yangtze Provinces focused in this study are Jiangsu (Administrative code CN032), Zhejiang (CN033), Anhui (CN034), Jiangxi (CN036), Hubei (CN042), and Hunan (CN043) (in Fig. 1). This defined MLYP provincial boundary would also constitute the region from which climate data will be extracted. Shanghai (CN031) is left out of the study not only because its jurisdictional area is a mega-city (larger than municipal-level cities but significantly smaller than provinces) but also due to its apparent inclination toward a financial-centric, non-agricultural economy.

Figure 1 Provincial-level administrative boundaries and hydrography of the Mid-Lower Yangtze Plains (MLYP) and surrounding areas.

MLYP provinces with a solid fill constitute this study’s area of interest.

Built environment.

The MLYP is densely populated, featuring multiple clusters of urban areas. In these urban areas, synergistic effects of increasing heatwaves and the urban heat island effect significantly worsen human thermal comfort and increase the demand for cooling (He et al., 2021). Addressing these localized climate impacts through improved urban planning, green infrastructure, and heat management strategies will be crucial for enhancing the resilience of cities and agricultural systems in this important region (He et al., 2022).

Natural environment.

The MLYP provinces have a dense network of rivers, natural lakes and reservoirs, and a temperate to subtropical climate with a significant winter-summer temperature difference (Tao et al., 2011). Amongst the provinces of interest, two (Jiangsu and Zhejiang) feature a coastline. The provinces are influenced by stable seasonal precipitation patterns dependent on frontogenesis triggered by interactions between the modified continental arctic and the tropical maritime air masses (Hou & Guan, 2013), such as the the June-July “Meiyu” (Zhang et al., 2015). Recently intensified intense convection weathers (Hu et al., 2021) and increasing heatwave occurrences (Mu & Huang, 2023) show that the MLYP is not spared from anthropogenic climate change.

Observation data

Referencing to the typical grain production function proposed by Just & Pope (1979), this study included three provincial-level variables: annual sown area of grain crops(x1: SownArea) number of primary-industry-employed personnel (x2: PrimInd, a measure of labor), and total power of agricultural machinery(x3: MechPower, a measure of physical capital). The number of years elapsed since the first recorded data point implemented in the study (x4: dt, a time dummy) is also included. dt improves the completeness of agricultural production functions economically as an input elasticities and factor productivity multiplier (Gong, 2018). It also boosts model performance in regression analyses (Just & Pope, 1979).

As shown in Table 1, SownArea, PrimInd, and MechPower are obtained from provincial and China Statistical Yearbooks produced by the National Bureau of Statistics of P. R. China (2022) and cross-checked with Eastern China Statistical Yearbooks from the Eastern China Statistical Information Network (1997). Recent editions of the said Yearbooks are obtained from the respective authorities. The Nanjing Library provides archived historical data.

Table 1 Observations.

(a) Variables considered	
Name	Unit	temporal Res	# of Cases
on t Axis1	
Total grain production	t	Annual	192	
SownArea2	kilohectare (kha)	Annual	192	
PrimInd3	thousands (*1000)	Annual	192	
MechPower4	gigawatts (GW)	Annual	192	
dt5	year	Annual	n/a	
tas6	K	Monthly	2952	
pr6	mm	Monthly	2952	
hw_freq7	year−1	Annual	246	
cw_freq8	year−1	Annual	246	
(b) Climate datasets utilized	
Dataset name	Variable	temporal Res	spatial Res	
GHCN_CAMS Gridded 2m Temperature (Land)a	tas	Monthly	0.5° × 0.5°	
CPC Global Unified Gauge-Based Analysis of Daily Precipitationa	pr	Monthly	0.5° × 0.5°	
CPC Global Unified Temperaturea	tasmax tasmin	Daily	0.5° × 0.5°	
Notes.

1 There is no data point missing. Data points for each province are evenly spaced on the time axis. Unless otherwise stated, the first year included is 1990 and last year 2021.

2 Actual sown area of grain crops.

3 Total number of personnel employed in the primary industry, a measure of labor.

4 End-of-year total power of agricultural machinery, a measure of physical capital.

5 Time elapsed since 1990, a generated variable following time increment.

6 First month: Jan. 1981, last month: Dec. 2021.

7 Heatwave (HW) frequency. First year: 1981, last year: 2021. Deduced from tasmax.

8 Coldwave (CW) frequency. First year: 1981, last year: 2021. Deduced from tasmin.

a Data provided by the NOAA PSL, Boulder, Colorado, USA, from their website at https://psl.noaa.gov.

Four variables are used as climate/meteorological inputs: mean air temperature (tas), measured in °C; total precipitation (pr), measured in mm; HW frequency (hw_freq); and CW frequency (cw_freq). As listed in Table 1B, the GHCN-CAMS Temperature dataset provided by NOAA PSL is used for tas, and pr data is obtained from the CPC Global Unified Gauge-Based Analysis of Daily Precipitation. “CPC Global Unified Temperature” supplies the daily minimum and maximum temperature values used to deduce hw_freq and cw_freq. To maintain cohesion on the temporal axis, the pr dataset is converted to monthly by summing each natural month. On the space axis, climate variables remain as grids.

GCMs

The SSPs outline several different pathways based on varying degrees of socio-economic challenges to mitigation and adaptation. Specifics for each SSP scenario are listed in Table 2. These scenarios range from sustainable development to regional rivalry and fragmentation (Bond & Scott, 2022). SSPs are integral to the ScenarioMIP activity of CMIP6, providing narratives that guide the projection of future emissions and land use in the models.

Table 2 Featured CMIP6 ScenarioMIP experiments.

ID	Activity	Experiment	Start - End	Description	
Historical	CMIP	All-forcing simulation of the recent past	1850 - 2014	CMIP6 historical.	
SSP126	ScenarioMIP	Update of RCP2.6 based on SSP1	2015 - 2100	Future scenario with low radiative forcing by the end of the century. Following approximately RCP2.6 global forcing pathway but with new forcing based on SSP1. Concentration-driven.	
SSP245	ScenarioMIP	Update of RCP4.5 based on SSP2	2015 - 2100	Future scenario with medium radiative forcing by the end of the century. Following approximately RCP4.5 global forcing pathway but with new forcing based on SSP2. Concentration-driven.	
SSP370	ScenarioMIP
AerChemMIP	Gap-filling scenario reaching 7.0degC based on SSP3	2015 - 2100	Future scenario with high radiative forcing by the end of the century. Reaches about 7.0 W/m2 by 2100; fills a gap in RCP forcing pathways between 6.0 and 8.5 W/m2. Concentration-driven.	
SSP585	ScenarioMIP	Update of RCP8.5 based on SSP5	2015 - 2100	Future scenario with high radiative forcing by the end of the century. Following approximately RCP8.5 global forcing pathway but with new forcing based on SSP5. Concentration-driven.	

CMIP6 models and projections used in this study are NASA’s NEX-GDDP CMIP6 dataset prepared by Thrasher et al. (2022). As part of the NASA Earth Exchange (NEX) project, it provides downscaled climate projections from the CMIP6 GCMs across all four “Tier 1” SSPs, making them suitable for regional climate studies. CMIP6 historical and ScenarioMIP experiments SSP126, SSP245, SSP370, and SSP585 of 25 Downscaled GCM models are selected for projecting future climate-induced grain output variability in the MLYP. This dataset is obtained from NASA’s NCCS THREDDS Data Server (NASA NCCS, 2024). More information about these GCMs can be found in Table S5.

Methodology

In adjacent fields, Zhao et al. (2022) developed a “Deep-Learning Autoencoder-Convoulutional Neural Network-Random Forest” algorithm for rapid and accurate prediction of soil texture, but few had developed similar approaches for grain production modeling. With the pipeline proposed in Fig. 2, this study developed a novel combination of two ConvAE architectures with a space-aware RF regression: the ConvAEs reduce the size of climate data from large grids to a multidimensional vector processable by regression algorithms. This enables the RF regression to recognize both spatial and temporal dynamics in climate variables. This section is dedicated to elucidating the study’s workflow part by part.

Figure 2 Proposed model pipeline that combines two types of convolutional autoencoders and a random forest regression serially.

Data preprocessing

The proposed model combines spatiotemporal information in reanalyzed meteorological datasets with time series of provincial economic variables to model MLYP grain output. Cases used to train the final model are grouped by year. For each case, gridded meteorological data is first normalized to a uniform resolution, then annual HW and CW frequencies are obtained. To deduce HW and CW frequencies, previously accepted definitions of ETEs, such as that featured in Perkins-Kirkpatrick et al. (2017) and Smith, Zaitchik & Gohlke (2012), are assessed, and, in this study, the 90th percentile of the cumulative density function (CDF) for each point on the 0.5° * 0.5° daily maximum temperature (tasmax) grid 1981-2010 would define the minimum HW-qualifying temperature of a specific location and a day will be labeled as ‘warm day’ when daily tasmax exceeds the threshold over at least three consecutive days. CW is determined following similar principles: a series of cold days on a point in the grid would be considered a coldwave if such a series contains strictly three or more consecutive days with tasmin lower than the 10th percentile value of the CDF for the point’s daily tasmin 1981–2010 data. Specifics regarding ETE definitions can be found in the author’s previous work (Mu & Huang, 2023).

The processed meteorological data are then encoded into eight codes with two distinct ConvAE architectures, each specialized for a specific temporal resolution. Meteorological data are then combined with the corresponding dataset for each economic variable normalized to the closed range [0,1]. Hence, right before the RF regression is performed, each case in the input dataset would contain four economic indices and 32 meteorological indices.

Convolutional autoencoders

Previous studies extensively used convolution, either in the form of a convolutional autoencoder (ConvAE) or a convolutional neural network, to facilitate dimension reduction. Hameed et al. (2022) applied a ConvAE-based deep learning approach for aerosol emission detection. Yu (2022) introduced a constrained dense ConvAE and deep-neural-network-based semi-supervised method to facilitate geological formation recognition. In this study, ConvAEs simplify the final RF regression and improve model performance by reducing potential noise.

Two ConvAE architectures are proposed (Fig. 2, Fig. S2). The spatial convolutional autoencoder (hereafter referred as the sAE) specializes in encoding 2d annual hw_freq and cw_freq datasets. Through two 2d convolution layers (each followed by a Rectified Linear Unit (ReLU) for zero activation) and dense layers, eight-dimension vectors are created as embeddings for each dataset. The spatio-temporal convolutional autoencoder (hereafter referred as the stAE) specialized in encoding 3D monthly tas and pr datasets. Through 3 3D convolution layers (also followed by a ReLU) and dense layers, eight-dimension vectors are created as embeddings for each dataset.

ConvAEs are trained in batches with an unsupervised algorithm based on how well the decoder can reconstruct the sample from the code encoded by the encoder. Mean squared error (MSE) is used to quantify the reconstruction error (i.e., act as the model performance criterion); a higher total MSE loss of the batch penalizes the current weights. Backpropagation computes the gradients. The Adaptive Moment Estimation (Adam) optimizer then uses the computed gradients to update model parameters. For more information on ConvAE training, please reference Supplementary Section S1.1.

Random forest regression

This study utilized the RF regression implementation created by Pedregosa et al. (2011). In RF regression tasks, each tree is constructed by recursively splitting the data based on feature values. Splits are chosen to minimize the variance within each node. Once all trees are grown, the RF regressor aggregates their predictions to form the final model output. In this study, this aggregation is the average of the predictions from all trees. This averaging mitigates overfitting and reduces model variance, leveraging the central limit theorem.

In this study, the RF input set (processed observed economic and climate data in the timeframe [1990,2021]) is split 80%-to-20% into training/validation and testing sets. A grid search optimization process is utilized to tune the four tree-growing hyperparameters of the RF regressor (n_estimators, max_depth, min_samples_split, min_samples_leaf). Additionally, one tree-pruning parameter, ccp_alpha, which quantifies the complexity parameter of the minimal cost-complexity pruning algorithm, is also optimized.

RF model evaluation

Mean absolute percentage error (MAPE), explained variance score (EVar), and the Gamma deviance (Dγ) score are used for evaluating the performance of the RF regression. MAPE and Dγ are used in the assessment of RF hyperparameter optimization success. MAPE assesses prediction error and model accuracy. MAPE is comparable across different scales as it quantifies error by percentage rather than absolute value. A lower MAPE indicates better model accuracy. EVar is used to quantify data spread. Generally, the explained variance score ranges from 0 to 1, where 1 indicates perfect prediction. EVar is also interpreted as the proportion of variance in the dependent variable that is predictable from the independent variables. By fitting them into a Gamma distribution, Gamma deviance (Dγ) quantifies the discrepancy between observed data and the values predicted by the model being evaluated. Mathematically, it is expressed as: Dγy,μ=2∑i=1nyi−μiμi− logyiμi.

Here, yi denotes the observed value and μi the expected value under the model for the i-th observation. The term log(yi/μi) arises from the log-likelihood of the saturated model, and (yi − μi)/μi represents the contribution of each observation to the log-likelihood (i.e., the fit) of the proposed Gamma distribution. The deviance is a measure of how well the model predictions µmatch the observed data y. A lower deviance indicates a better fit of the evaluated model to the data.

Analyzing the impact of future climate change

Though SSPs contain data regarding land use and agricultural/industrial development, the resolutions of these datasets are not nearly high enough to enable reliable regional studies: GDP and employment projections are included at a national level (Bond & Scott, 2022), and the critical capital factor, power of agricultural machines, is not included in the scenarios. Hence, this study will attempt to evaluate future production possibilities under the current (2021) economic situation, and isolate the effect of climate change for analysis.

Model benchmarking

Multilinear regression.

Performance metrics of the proposed model is compared to a benchmark model based on a log–log multivariate regression model with parameters optimized using the Feasible general least squares (FGLS) approach implemented by Holst, Yu & Grün (2013) (the model is hereafter referred to as B1-FGLS). B1-FGLS is a lower-complexity attempt to describe grain production as an economic activity. Since Jeong et al. (2016) has proven RF regression to be superior to multilinear regression in grain production modeling, B1-FGLS also constitutes the expected baseline performance of the proposed ConvAE-RF model (which still utilized RF as the regression method). Implementation of this model can be found in Mu (2024). B1-FGLS is trained with the same observation dataset used for the proposed model. Spatial means are taken for each climate variable.

Ablation test.

This study also attempts to confirm the superiority of a spatio-temporal model with an ablation experiment featuring a RF model that disregards spatial characteristics of climate variables (B2-RF). The regression part of this algorithm uses the spatial means for each meteorological variable. RF Hyperparameters in the B2-RF model are optimized using the exact dataset and the same search grid as the proposed model. Through this approach, B2-RF quantifies the performance difference induced by awareness of spatial dynamics in climate variables.

Results

Projected ETE in MLYP

Before modeling, all climate datasets are pre-processed spatio-temporally. To utilize ScenarioMIP projections of future tasmin and tasmax to deduce ETE frequencies, the thresholds are downscaled linearly to a 0.25° * 0.25° grid, matching the spatial resolution of the NEX-GDDP-CMIP6 GCMs. Then, HW frequencies and CW frequencies are deduced according to the definitions outlined in ‘Data Preprocessing’.

Spatial patterns

Figure 3 shows the spatial distribution of ETEs across three 20-year periods and 2 SSP scenarios. The two extremes differ spatially: HW frequency increase is more significant toward the equator and insignificant in higher latitudes, while CW frequency consistently declines across space–time. Currently, as in the upper left panel of Fig. 3, higher mean annual HW frequencies are observed in the North of the Yangtze river than in the South. Mean HW frequency predicted by selected GCMs for the provinces to the Northern side of the MLYP shows (1) a less significant increase of HW frequency over time and (2) less experiment difference between different SSP scenarios. Among the MLYP provinces, over the last 20 years, Anhui and Zhejiang generally have the highest number of mean HW_freq. However, Anhui’s mean annual heatwave frequency may slightly decrease in the late 21st century, as reflected by heatwave frequency’s Pearson correlation with time graphed in the upper right panels of Fig. 3.

Figure 3 Map of CMIP6 GCM projected ETE frequencies in the MLYP provinces, including current observations, SSP245 temporal range means, SSP585 temporal range means, and temporal Pearson correlations plotted for each point on the spatial grid.

SSP126 and SSP370 temporal range means can be found in Fig. S6.

Some discontinuities from observed HW frequency are observed across the spatial distribution of HW frequencies in SSP scenarios. HW hotspots in Northern and Eastern Zhejiang provinces and spots of reduced HW frequencies surrounding major reservoirs in Hunan, Anhui, and Hubei are seen across a few of the 25 GCMs. There is also slight projected decrease in the frequencies of HW occurrence in historically warm regions around MLYP, especially in coastal Zhejiang and Fujian (out of scope).

Temporal patterns

Processed terrestrial HW and CW frequencies are averaged to 4-year means across the MLYP provinces. SSP scenarios all project a general increase in spatial mean HW frequency in Fig. 4A. But, starting with the 2057-2060 group, the SSP370 ensemble is beginning to project a slightly higher number of median HW frequencies than SSP585 (while both still fall under a similar range). This surpassing will be more significant at the end of the century: both the median and maximum of SSP370 ensemble heatwave projections would be approximately 10% more than that of SSP585. This is because of the projected land use changes innate in SSPs (Bond & Scott, 2022). Another observation: the SSP126 scenario projects a much higher level of CW frequency in 2061-2100 than SSP245 (max median difference reaching 2 at 2097-2100, max max difference reaching 1 at 2061-2064). Although 4-year medians and ranges of HW frequency of their groups are relatively similar throughout the projected timeframe, with a numerical max maximum difference merely reaching +1, which did not come until 2097-2100, max of SSP126 annual mean HW frequency projection has also exceeded that of SSP245 in several year groups.

Figure 4 (A) Distribution of point-level projected HW frequency in the MLYP aggregated over 4-year-groups vs. time, by SSP scenario. (B) Distribution of point-level projected CW frequency in the MLYP aggregated over 4-year-groups vs. time, by SSP scenario. (C) Distribution and linear correlation of flattened spatio-temporal gridded HW projections and CW projections, by SSP scenario.

According to Fig. 4B, CW frequency has a stronger negative relationship with time than HW frequencies, especially in scenarios with higher radiative forcings. In SSP585, by the end of the century, more than 50% of the points projects less than two CW events per year. Figure 3 supports this trend. The extent of such decrease does not vary significantly spatially, but there is a clear trend across time. The higher increase in temperature projected by SSP585 made its mean projected CW frequency significantly lower, and Pearson correlation results of SSP585′s projected CW frequency over time have an especially large magnitude that almost doubles that of SSP245′s. Noteworthy temporal trends appear in HW frequency’s temporal projections.

HW-CW relationship

The trade-off between HWs and CWs in the rest of the 21st century is plotted Fig. 4C. Via a stepwise regression analysis, it is determined that CW frequency has a significantly lower influence on HW frequency than the reverse. In Fig. 4C, reducing CW frequency by 1 year−1 increases HW frequency by 0.25 year−1, and conversely, increasing HW frequency by 1 year−1 is matched by an approximately -4 year−1 change in CW frequency.

Current grain cultivation in MLYP

In the past decades, Chinese agriculture experienced a transformation from labor-intensive agriculture to industrial agriculture that relies more on machines. As in Fig. 5, this shift from labor- to machine-based agriculture is well visible in the base data implemented in this study: while a high number of personnel are employed in the primary industry in the last decades of the 20th century, none of the MLYP provinces have more than two million workers in agriculture after 2005. Similarly, an increase in the wealth of rural households and a boom of industrialized agriculture contributed to a significant rise in agricultural machinery (Xie & Jin, 2015): total power of agricultural machinery more than tripled in most MLYP provinces over 32 years (1990–2021).

Figure 5 Observations of (A) actual sown area of grain crops, in thousand hectares (kha), (B) end-of-year employed personnel in the primary sector, in thousands, and (C) end-of-year total power of agricultural machinery in the MLYP provinces, in Gigawatts (GW).

A linear trend line is created for each province in (B) and (C).

Meteorological data are passed through ConvAEs, and the outputs are compiled together with the four economic variables into 36-dim vectors to form the input dataset of the RF regression. The RF’s parameters are optimized in a search grid (Fig. 6, refer to Section S1.1 for more info). Table 3 shows the importance of each variable in the response. SownArea plays the most crucial role in determining total grain output. This is followed by tas. cw_freq (CW frequency) comes third. The aggregated influence of the eight hwave codes and the eight cwave codes are significantly higher than that of PrimInd, which has a similar level of importance as dt and pr. Observed and predicted total grain production for each province over time are included in Fig. S4.

Figure 6 RF regression (A) k-fold cross-validation criterion (MAPE) and (B) model performance criterion (Dγ) over combinations of n_estimators (Total Number of Trees) and max_depth (Maximum Tree Depth) searched in the hyperparameter optimization process.

Other hyperparameters have been optimized. The vertical line denotes selected n_estimators and max_depth.

Table 3 RF regression variable importance (MDI).

Variable	Importance	
SownArea	0.63356	
dt	0.02186	
PrimInd	0.02109	
MechPower	0.05399	
tas	0.12350	
pr	0.02023	
hw_freq	0.05008	
cw_freq	0.07567	

Model evaluation

The three model evaluation statistics are calculated for both the proposed ConvAE-RF and benchmark models B1-FGLS and B2-RF (Table 4). All three evaluation metrics are used for comparison with the benchmark models. The proposed model has the lowest MAPE (49.2% of B1-FGLS and 59.9% of B2-RF), the highest amount of variance captured (112.0% of B1-FGLS and 105.0% of B2-RF), and the lowest Dγ (19.5% of B1-FGLS and 35.0% of B2-RF, therefore a higher model likelihood) among the three models. Based on the three metrics, the optimized ConvAE-RF model demonstrates superior performance over the multilinear grain production model and the common spatial Random Forest regression model in the MLYP. The second best among the three evaluated models is the common temporal RF regression model. Both RF-based models performed significantly better than the multilinear baseline (B1-FGLS), which aligns with previous research. The results are listed side-by-side in Table S3.

Table 4 Evaluation statistics of the proposed model (bold) and two benchmark models.

Model	MAPE value (% of ConvAE-RF)	Evar value (% of ConvAE-RF)	Dγ value (% of ConvAE-RF)	
B1-FGLS	7.367e−2 (203.2%)	0.8721 (89.3%)	1.106e−2 (512.7%)	
B2-RF	6.052e−2 (166.9%)	0.9291 (95.2%)	6.168e−3 (285.9%)	
ConvAE-RF	3.626e−2	0.9764	2.157e−3	
Notes.

The row containing data for the proposed model is highlighted in bold for emphasis.

Future grain production in MLYP

Economic data are kept constant at the current level. NEX-GDDP-CMIP6 GCM data are encoded via optimized ConvAEs for the respective variables. Each GCM is passed through the RF, and the median predictions are compiled in Fig. 7A. The mean and mid-50% bounds of current, mid-century, and end-of-century output projections are recorded in Table S4. Ceteris paribus, following model predictions from the tier-1 ScenarioMIP activities, by the 2050s, the median GCM projects grain output to decrease by roughly 107 tons (−0.736% from 2020s) for SSP1, 66 tons (−0.451% from 2020s) for SSP2, 83 tons (−0.568% from 2020s) for SSP3, and 154 tons (−1.055% from 2020s) for SSP5. Compared to RCP-based by-crop studies, this study projects a lower %-decrease in overall grain production up to the 2050s. Specifically, output risk at SSP2 is close to 80% less than that projected for rice by Chen et al. (2020). From the 2050s to the end of the century, grain production would be approximately constant in SSP1, decreasing around 41 tons (107 tons from 2020s, −0.733%) for SSP2, 12 tons (95 tons from 2020s, −0.651%) for SSP3 and 116 tons (270 tons from 2020s, −1.848%) for SSP5. Risk-wise, in the SSP585 scenario, over 50% of models project a lower total grain output in the MLYP provinces than the baseline plotted in Fig. 7A, which is the mean total grain production in MLYP in the first 20 years of the 21st century.

Figure 7 Median and range of projected MLYP annual grain production under current economic conditions and SownArea using (A) data from 25 ScenarioMIP models, (B) data from models whose hw_freq projection is above the 50th percentile of the 25 models, and (C) data from models whose hw_freq projection is below the 50th percentile of the 25 models.

Output projections from 2021-2100 are smoothed over four years. The range is interquartile. The black dotted line is MLYP’s mean actual grain production from 2001-2020.

The GCMs can be sorted based on the sum of their point-wise differences based on projected HW frequency (Table S6). Generally, similarities and differences in their AOGCM components impact HW projections of GCMs systemically. The sorted GCMs are then split into two ensembles: the “higher” if their HW frequency projections fall into the top 50%, and the “lower” if their HW frequency projections fall into the bottom 50%. The specific order can be found in Table S6. Figures 7B and 7C split Fig. 7A by GCMs into “lower” and “higher” ensembles according this definition.

Comparing the two ensembles, those projecting higher HW frequencies initially project higher annual grain output. However, as highlighted in ‘HW-CW Relationship’, the more significant decrease in CWs corresponding to higher HW frequencies should be held more accountable for this increase. Moreover, across scenarios, a significantly higher level of output volatility is seen between 2021 and 2060 in the “higher” ensemble, with fluctuations of the 4-year median output of as much as over 150 tons (seen in SSP585 2055-2058 anomaly from 2051-2054). Max output fluctuations in the “lower” ensemble in that same time range is at just over 110 tons, more than 30% less than the “higher” ensemble.

Concerning scenarios, SSP245 grain production projections in the lower ensemble (Fig. 7B) in the second half of the 21st century fluctuated around 14,500 tons and never broke the baseline, but SSP245 in the higher ensemble quickly broke the baseline at 2071-2074, and sustained there, 50 tons below the value the lower ensemble settled around. In the highest radiative forcing scenario, SSP585, the median of MLYP grain output projections by the higher ensemble dives below baseline (and sustains there) as early as the mid-2060s. In comparison, the lower ensemble maintained until the mid-2070s and broke the baseline. Interestingly, starting in the 2060s, with nonstop fluctuations, 4-year median output projection of the SSP370 scenario in the higher ensemble exceeds the output projected of SSP245, and in two 4-year smoothing periods exceeded that of the sustainability scenario, SSP126. However, the median and range of the SSP370 projection do not platform nor slow down in its decrease; instead, regression analysis on the medians shows an increasing slope over time. This indicates that optimistic projections of SSP370 grain output should be subject to further investigation.

Fortunately, as seen across the two ensembles, if sustainable development and land use are achieved, neither of the ensembles projects a substantial decrease in grain output by the end of the century, which, again, is a result explicable by the extra reduction of CWs that neutralized the impact of some heatwaves.

Impact of average temperature change on grain production.

The tas anomaly from the 1981-2010 climatology projected by each GCM are drawn to grain production anomaly in Fig. 8. Adhering to the Central Limit Theorem, anomalies are averaged across each period of interest. Regression analysis excluding outliers shows a clear negative relationship between grain production and tas anomaly across scenarios in the following decades, mid-century, and end-of-century MLYP grain production projections, with regression certainty increasing over time. However, step-wise regression analysis on 2021-2040 points to a roughly 200 tons (1.67%) decrease in grain production with a one degC increase. On 2041-2060, step-wise regression points to an approximately 145 tons (1%) decrease in grain production with a one degC increase. In 2081-2100, grain production decreased approximately 120 tons with a one degC increase in mean tas. Despite the slight, this decrease in response magnitude is noteworthy.

Figure 8 Change in mean air temperature (tas) projected by 25 selected CMIP6 models on periods 2021-2040, 2041-2060, and 2081-2100 under SSP126, SSP245, SSP370, SSP585 scenarios relative to baseline in the MLYP.

The dotted line is roughly the minimum mean tas anomaly for SSP585.

Discussion

Model performance

According to Table 4, the optimized ConvAE-RF model performs significantly better than the optimized B1-FGLS baseline in all three evaluated metrics. This result aligns with Jeong et al. (2016) and Fradgley et al. (2023), indicating successful model optimization efforts.

The optimized ConvAE-RF model is also more accurate (lower MAPE) and precise (lower EVar) than B2-RF. With the lowest Dγ amongst the three evaluated models, the proposed model is the best fit for the MLYP dataset. It would be safe to conclude that the space-aware RF regression powered by convolutional embeddings performs superior to the common RF regression in the modeling of grain production in the MLYP. This indicates that added spatial complexity can increase the performance of a predictive RF model, which aligns with the observations about RF-based models in adjacent fields, such as Zhao et al. (2022).

However, it should be acknowledged that increased model complexity is making it harder to understand and interpret the results. The black-box nature of neural networks, coupled with the ensemble nature of random forests, complicates the interpretability further. Therefore, discussion of results will put a heavier focus on regional patterns over province-specific results, which may be inaccuarate due to a range of influences (more about such influence in ‘Limitations and Uncertainties’). Additionally, the model performs well in the mid-lower Yangtze plains (MLYP), but its applicability to other regions cannot be guaranteed.

Projected ETEs and their negative influence

In the proposed model, climate-induced decline in grain production in the MLYP is comprised of two general stages: decline and platform. Up to the 2070s, MLYP grain production is projected to decline steadily (with step-wise decline increasing as Scenarios escalate from SSP126 to SSP585) (Fig. 7). The diminishing influence of overall climate change is explained by the mixed influence of ETEs at higher temperature anomalies. This is especially significant in high-radiative-forcing scenarios of SSP370 and SSP585, where CW frequency diminishes significantly as a tradeoff for increased HW. At this stage, production decline from the damaging factors is offset by decreasing CW frequency’s relatively impactful positive influence. Production decrease in the SSP585 scenario lies within the 0%–4% range projected with RCP8.5 by Sun et al. (2023).

Historically, extreme temperatures impact more on grain production than average temperature (Liu et al., 2022). This study supports this conclusion in a timeframe beyond the status quo. From the RF regression, hw_freq’s variable importance on MLYP grain production is 237.4% that of PrimInd, 247.5% that of pr, and 40.5% that of tas; cw_freq’s variable importance on MLYP grain production is 358.7% that of PrimInd, 374.0% that of pr, and 61.3% that of tas (Table 3).

Laboratory studies, such as Rodríguez et al. (2015) and Gao, Tester & Julkowska (2020), indicate that heat stress (appearing in nature as HW events) reduces rates of photosynthetic efficiency and fresh weight observed in grain crops upon exposure in lifetime. Combined with its considerable impact to the performance of the ConvAE-RF model (Table 3), HW frequency can be held fairly accountable for the decrease in grain production. CWs are also capable of hindering rice production (Kim et al., 2021) and negatively impact the quality of crops (Mathivanan, 2021), so significantly decreasing CW frequency over the rest of the 21st century across scenarios (Fig. 4) may be a remedy.

Comparing Figs. 7B and 7C, despite the absence of a solid median output difference, the projected median grain outputs of all scenarios in the higher ensemble are much more volatile than the lower ensemble. Moreover, in the higher ensemble, median curves in SSP126, SSP245, and SSP585 almost mirror the shape of HWs and CWs. This resemblance implies a causality relation, albeit a holistic causality test was not performed due to complexity constraints.

Impact of changing temperatures

RF regression output in Table 3 places mean air temperature (tas) as the 2nd most crucial variable (after total sown area of grain crops, SownArea). In other words, tas significantly influences the model’s performance, and changes in tas are highly associated with changes in MLYP grain production. Number-wise, as in Fig. 8, grain production variability with tas anomaly is most uncertain when tas anomaly is in the range of 2 degC to 3 degC. When tas anomaly is above six degC, none of the models project an increase in grain production from baseline. As in Fig. 7, decreasing production projections under a warming climate corroborates this assessment. However, it should be noted that the influence of temperature change on production in this study seem to be slightly lower than the conclusions of the previous studies, such as Chen et al. (2020) and Holst, Yu & Grün (2013).

Drawing the variability in grain output over time by province provides a plausible explanation (Fig. 9). CW frequency decrease in Jiangxi and Jiangsu and HW & tas increase in northern Anhui (‘Spatial Patterns’) provided more favorable conditions in previously low-temperature crop growing periods, thereby offsetting some negative impacts of climate change in the broader MLYP zone. Province-level output variability (Fig. 9) also warns against significant, negative consequences in high-radiative forcing scenarios, with multiple provinces expected to incur a grain output decline larger 1% in the 20-year periods after 2060. More climate-induced grain production variations is also projected across provinces in higher-radiative-forcing scenarios: a climate-induced provincial grain production fluctuation greater than 1%, whether negative or positive, is never projected in SSP126 (Fig. 9A) or SSP245 (Fig. 9B). For why positive changes may occur (especially Anhui and Jiangxi), please refer to the trade-off between HWs and CWs discussed in ‘Projected ETEs and Their Negative Influence’.

Figure 9 Mean projected annual grain output in 2020-2040 and following 20-year changes for each province averaged across 25 selected CMIP6 climate models in the (A) SSP126 (B) SSP245 (C) SSP370 (D) SSP585 scenarios.

The existing literature does not reach a general agreement on pr’s influence on grain output, regional studies (such as Sharma et al. (2019)) and studies that included the MLYP as a significant part of a specific sector under bigger scales (such as Holst, Yu & Grün (2013)) agreed on the slight negative influence of increasing precipitation in the temperate zone on grain production and a strong negative impact of extreme precipitation (Liu et al., 2022). Yue et al. (2021) claims that precipitation in the future may increase. This provides for a possibility that may participate in neutralizing the negative influence of tas, especially when torrential rains are considered.

Limitations and uncertainties

Due to the high complexity of this model and the diversity of our data sources, there are a number of noteworthy uncertainties in this study.

Model-wise, the combined use of the ConvAE architecture to handle spatial complexity and the RF regression to handle inter-variable relationships has helped this study to perform superior to existing models, as mentioned in ‘Model Performance’. However, the proposed model requires a substantial amount of data and may be less scalable. This could be a limitation to applying the proposed to larger or atypical datasets in different geographical areas and climates.

Data-wise, future climate projections comes from GCMs. While GCM data provides valuable insights into climate patterns, it is subject to inherent uncertainties due to model assumptions, resolution limitations, and the representation of complex physical processes (Thrasher et al., 2022). Consequently, predictions based on GCM data should be interpreted with caution, considering the potential variability and the influence of factors not fully captured in the models. A discussion of how the GCMs in this study exhibit such variations can be found in the Supplementary Section S2. There is also a level of difference in crop type, plant phenotype, and farming techniques across regions. Within the area of study, Anhui and Hubei are the only provinces that predominantly cultivate a single kind of crop. Anhui is a major producer of Paeonia ostii (Peng et al., 2017), while Hubei Oryza sativa (Fan et al., 2018).

External complexities.

The influence of sociopolitical role changes on province behavior and non-climate catastrophes on production possibilities should be recognized. Supply-side reforms in Anhui (Government of Anhui Province, 2020), for example, increased the total grain output in Anhui in the short term (which, in Fig. S4, the proposed model did not simulate accurately). The COVID-19 catastrophe of 2019–2020 also reduced grain production output in Jiangxi, Hubei, and Hunan Province (Li, 2020). During this period, Hunan province experienced an extreme decrease in the expected grain output. Provided that the proposed RF model has taken into account SownArea, PrimInd, and MechPower (which all did not have a substantial change over the period), the likelihood for error is relatively low, and it would be explicable if resorted to the influence of COVID-19, which can also explain the decline in PrimInd (personnel employed in the primary sector) in 2020. Moreover, in Jiangxi and Hunan, output anomaly fell below zero after 2015, but according to Hou (2018), recent supply-side reforms swapped a large number of old phenotypes with new phenotypes, thereby remedying output decrease (Fig. S4).

Conclusion

This study presents a novel approach to modeling grain production in the Middle-Lower Yangtze Plains (MLYP) region of China, combining convolutional autoencoders (ConvAEs) and random forest (RF) regression to capture the spatial and temporal dynamics of climate variables. The proposed ConvAE-RF model outperforms benchmark models in terms of accuracy, precision, and goodness-of-fit, demonstrating the value of incorporating spatial complexity into predictive models.

The analysis of projected extreme temperature events (ETEs) reveals concerning trends. Heatwave (HW) frequency is projected to increase significantly, especially in high-radiative-forcing scenarios like SSP370 and SSP585, while cold wave (CW) frequency is expected to decrease. The trade-off between HWs and CWs shows that reducing CW frequency by 1 event per year can increase HW frequency by 0.25 events per year, while the reverse relationship is much stronger, with a 1 event per year increase in HW frequency corresponding to a four event per year decrease in CW frequency. These projected changes in ETEs are expected to have a substantial negative impact on grain production in the MLYP region. The model projects a climate-induced decline in grain production up to the 2070s for all scenarios, with the rate of decline increasing as the radiative forcing scenario escalates from SSP126 to SSP585. This decline is primarily driven by the damaging effects of increased HW frequency and increased daily average temperatures. After 2060, grain output variability is significantly determined by the HW-CW trade-off. Therefore, the lower radiative forcing scenarios SSP126 and SSP245 project more stable grain production than SSP370 and SSP585.

The findings of this study have important implications for policymakers and agricultural stakeholders in the MLYP region. The insights into the spatial and temporal dynamics of ETEs and their impact on grain production can inform adaptation strategies, such as the development of heat-resistant crop varieties, improved irrigation systems, and the implementation of early warning systems. Additionally, the successful application of the ConvAE-RF model demonstrates the potential of leveraging advanced machine learning techniques to enhance agricultural forecasting and decision-making in the face of climate change.

While there is a projected decline in grain production due to climate change, more sustainable development and land use practices could mitigate some of the negative impacts. However, the specific effects on different types of grain crops and the potential for new crop phenotypes to alter these projections require further research. A deeper dive into adaptive mechanisms can also buffer against the adverse impacts of temperature extremes, possibly leveraging genomics-assisted breeding and other innovative techniques.

Supplemental Information

Supplemental Information 1 Supplementary Materials, Tables and Figures.

The authors would like to acknowledge the Nanjing Library for providing access to its archived series of Chinese Statistical Yearbooks, Eastern China Statistical Yearbooks, and Province-level Statistical Yearbooks. The authors would like to acknowledge the United Nations Office for the Coordination of Humanitarian Affairs (OCHA) for providing base maps to Figs. 1, 3, 9, and Fig. S6.

Abbreviations

B1-FGLS Benchmark model 1: the baseline log–log multilinear regression model optimized with Feasible Generalized Least Squares, used as a performance baseline

B2-RF Benchmark model 2: the common temporal Random Forest regression model used in ablation test

CMIP5 Coupled Model Intercomparison Project Phase 5

CMIP6 Coupled Model Intercomparison Project Phase 6

ConvAE Convolutional Autoencoder

CW Coldwave

D γ Gamma Deviance Statistic

ETE Extreme Temperature Event(s)

EVar Explained Variance

FGLS Feasible General Least Squares

GCM General Circulation Model

GDP Gross Domestic Product

HW Heatwave

MAPE Mean Absolute Percentage Error

MechPower End-of-year total power of agricultural machinery

MLYP Middle-Lower Yangtze Plains

PrimInd Total number of personnel employed in the primary industry

RCP Representative Concentration Pathway

ReLU Rectified Linear Unit

RF Random Forest

sAE the proposed spatial Convolutional Autoencoder

ScenarioMIP Scenario Model Intercomparison Project (part of CMIP6)

SownArea Actual sown area of grain crop

SSP Shared Socioeconomic Pathway

stAE the roposed spatio-temporal Convolutional Autoencoder

YRB Yangtze River Basin

Additional Information and Declarations

Competing Interests

Author Contributions

Data Availability

Zijun Mu is employed by Nanjing Smardaten Technologies Co., Ltd. All other authors declare that they have no competing interests.

Zijun Mu conceived and designed the experiments, performed the experiments, analyzed the data, prepared figures and/or tables, authored or reviewed drafts of the article, and approved the final draft.

Junfei Xia conceived and designed the experiments, authored or reviewed drafts of the article, and approved the final draft.

The following information was supplied regarding data availability:

The data is available at Zenodo: Mu, Z. (2024). Datasets generated in the ConvAE-RF modelling of grain yield in the mid-lower Yangtze plains [Data set]. Zenodo. https://doi.org/10.5281/zenodo.10924805

The code (latest release) is available at Zenodo: Zijun Mu. (2024). zjmagou/MLYPGrain2024: 1.0 (1.0). Zenodo. https://doi.org/10.5281/zenodo.13359864.

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
