# Peer review of "Predicting the influence of extreme temperatures on grain production in the Middle-Lower Yangtze Plains using a spatially-aware deep learning model"

_PeerJ, doi:10.7717/peerj.18234_

## Round 0.1 · original submission · Major Revisions

We are pleased to inform you that your manuscript has been well reviewed, but needs some revisions before publication.

Reviewer 1 ·

Basic reporting

This paper is overall well written and the language is fine. Authors generally follow a clear logic.

Experimental design

It is overall good, and the analysis is mostly solid.

Validity of the findings

The Results are clearly stated.

Additional comments

The manuscript employs a convolutional autoencoder and random forest regression to model grain yields in the middle and lower plains provinces of the Yangtze River. The analysis considers socio-economic and meteorological factors in order to identify the factors influencing agricultural production. On this basis, the study proceeds to incorporate scenario development analyses, with a view to developing projections and assessing the impacts of climate extremes. The study commences with a robust foundation, a lucid rationale, an apt methodology, and a well-structured framework. Upon perusing the manuscript, I have the following suggestions and comments:

1. The introduction section employs a considerable amount of content to illustrate the advancement of the analysis methodology and the prediction model. However, there is a paucity of content that expresses the innovativeness and significance of this study.
2. Authors should present the impact of extreme heat challenges comprehensively. Please refer Localized synergies between heat waves and urban heat islands: Implications on human thermal comfort and urban heat management. Environmental Research, 193, 110584. Beating the urban heat: Situation, background, impacts and the way forward in China. Renewable and Sustainable Energy Reviews, 161, 112350.
3. The study proposes a spatially-aware deep learning model and its differences and advantages over the commonly used Random Forest (or geographically-weighted Random Forest model) should be reflected in the manuscript.
4. The assertion made in Line 107-108 is erroneous.
5. It is recommended that the Methods section be reorganised to eliminate well-known elements (e.g. definitions and formulae in section 3.3.1) and enhance the overall content of the section (e.g. sections 3.3, 3.4 and 3.5).
6. The study mentions multiple pathways for this predictive model CMIP6 in terms of data, but only two of them are selected in the final section 4.1.1. It would be beneficial for the authors to explain why these two pathways were selected over the others.
7. The authors present a substantial number of statements pertaining to provinces and regions in section 4.1.1. It is therefore recommended that the manuscript be augmented with statistical charts or tables based on provinces or regions, thus affording readers the opportunity to gain a deeper understanding of the subject matter.
8. The corresponding units should be added to the parameters in figure I.
9. Line 303 presents a discrepancy between the data and the presentation. Table 5 indicates that the cold wave may be more significant than mechpower.

In summary, it is recommended that the manuscript be given a major revision.

·

Basic reporting

Main comments:
This study investigates the response of crop yields in the Middle-Lower Yangtze Plains to extreme temperatures using meteorological products and statistics. A deep learning framework is proposed to elucidate critical environmental drivers and enhance yield prediction accuracy. The manuscript is well-structured and provides valuable insights. However, it contains numerous lengthy sentences that are challenging to follow and non-academic terminology that diminishes its rigour. This manuscript's text may benefit from professional language editing. Before publishing, major revisions are necessary to improve the manuscript's scientific rigour and completeness. Specific comments are as follows:

Experimental design

Introduction:
L15-17: Please go directly to the research background and remove unnecessary information.
L18-21: Clearly state the research objectives and the problems to be addressed.
L22: The unit for yield should not correspond to tons. Please correct it.
L29: 'Yield volatility' is an unusual term in the agricultural context. Please use an alternative expression.
L37: Specify whether the "influence on crop phenology" refers to the duration or beginning time.
L37: It is better to end the first paragraph with a statement on the significance or necessity, such as the importance of crop yield prediction under increasing extreme climate conditions.
L41: References are needed. Recommend citing article which provide applications of random forest methods in crop yield prediction. The reference to Jeong et al. (2016) is somewhat outdated.
L52: Change “to project future yield” to “to project future yield variations/losses.”
L57-59: The content is somewhat common knowledge. It is recommended that the literature review and research gaps be focused more on.
L64-65: Provide some statistical data to illustrate the impact of extreme weather on crop production in the Yangtze River Basin.
L69: When SSP is mentioned for the first time, please provide the full term.

Methods:
L95: Which crops are included in the calculation of grain production? Are the same crops considered across all provinces? Please provide details.
L98-100: Provide references or justification for why the variable x4 is appropriate.
L96-99: Provide more details on these statistical data, such as the time range, the existence of missing data, and reasons for selecting these variables over others.
L108: Are the variables in Table 1b averaged at the provincial level?
L127: How is “a space-aware” understood in this study? Please provide an explanation or elaboration.
There are numerous abbreviations in the paper that have not been defined, especially in the Data and Methods section (such as sAE, stAE, and DLAC-CNN-RF). Please provide the full terms when they first appear. Additionally, if an abbreviation is only used once, use the full term instead.
Figure 2 is difficult to understand. A detailed explanation of the specific workflow needs to be provided at the beginning of the Methods section.
The Convolutional Autoencoder (ConvAE) architectures need to be described in detail.
'Total yield' should be expressed as 'production'.
L169-174: This section overlaps with the Introduction SECTION.
How are the parameters for RF regression set in this study? Please provide specific measurements.
L229: What does FGLS mean?
Please provide a justification for using the benchmark model.

Validity of the findings

Results and Figures:
Abbreviations in Figure 2 should be defined in the captions, as should those in other figures.
In Figure 3, please provide numbering and indicate which are HW and CW.
Figure 4c may not have significant meaning in regressing two meteorological indicators.
Scientific notation in Table 3 makes it difficult for results to be compared.
Units should be provided in Figure 7.
L337-338: What are the criteria for defining the "higher" and "lower" sets?
L362: While this study focuses on average temperature changes, it is important to note that the impact of climate change extends beyond temperature alone.
Authors may need to provide more analysis to elucidate how extreme temperatures affect crop production. It is suggested that historical data be analyzed first, followed by a discussion of future changes.

Additional comments

Discussion and Conclusion:
Please discuss the strengths and limitations of the methods used in this study and compare them with other approaches.
L399: It would be ideal to discuss the relationship between crop productivity and climate change for each province.
Section 5.3 focuses solely on variations in HW prediction due to GCM differences, with limited discussion on yield. Authors should expand on how GCMs contribute to variations in yield.
Authors should also address uncertainties related to the study's data, methods, and models.
The conclusion section is overly lengthy. Please streamline it.

Reviewer 3 ·

Basic reporting

1. Please avoid to use abbreviation in the abstract
2. language need to improve in abstract section
3. Explain about the models and significance of
Spatially-Aware Deep Learning Model

Experimental design

1. Expand the GCMs

Validity of the findings

1. Discussion need refinement
2. do not repeat your result in discussion section
3. discuss your output in discussion with suitable references
4. conclusion should be summarized in brief

---

## Round 0.2 · accepted · Accept

This revised version is suitable for publication in PeerJ.

Reviewer 1 ·

Basic reporting

This paper showcases a clear structure, proper English, and sufficient background.

Experimental design

The research questions are well defined and the methods adopted are reliable.

Validity of the findings

Overall, the findings are solid and reliable for consideration.